



# Evaluation of carbonyl sulfide biosphere exchange in the Simple Biosphere Model (SiB4)

Linda M.J. Kooijmans[1], Ara Cho[1], Jin Ma[2], Aleya Kaushik[3,4], Katherine D. Haynes[5], Ian Baker[5], Ingrid T. Luijkx[1], Mathijs Groenink[1], Wouter Peters[1,6], John Miller[4], Joseph A. Berry[7], Jerome Ogée [8], Laura K. Meredith[9], Wu Sun[7], Kukka-Maaria Kohonen[10], Timo Vesala[10,11,12], Ivan Mammarella[10], Huilin Chen[6], Felix M. Spielmann[13], Georg Wohlfahrt[13], Max Berkelhammer[14], Mary E. Whelan[15], Kadmiel Maseyk[16], Ulli Seibt[17] , Roisin Commane[18], Richard Wehr[19,20], Maarten Krol[1,2]

[1] Meteorology and Air Quality, Wageningen University and Research, The Netherlands

[2] Institute for Marine and Atmospheric Research, Utrecht University, The Netherlands

[3] Cooperative Institute for Research in Environmental Sciences, University of Colorado Boulder, CO, USA

[4] NOAA Global Monitoring Laboratory, Boulder, CO, USA

[5] Department of Atmospheric Science, Colorado State University, USA.

[6] Centre for Isotope Research, University of Groningen, Groningen, The Netherlands

[7] Department of Global Ecology, Carnegie Institution for Science, Stanford,
CA, USA

[8] INRAE, Bordeaux Science Agro, UMR 1391 ISPA, 33140 Villenave d'Ornon, France

[9] School of Natural Resources and the Environment, University of Arizona, Tucson, AZ 85721, USA

[10] Institute for Atmospheric and Earth System Research/ Physics, Faculty of Science, University of Helsinki, Helsinki, Finland

[11] Institute for Atmospheric and Earth System Research/ Forest Sciences, University of Helsinki, Helsinki, Finland

[12] Yugra State University, 628012, Khanty-Mansiysk, Russia

[13] Department of Ecology, University of Innsbruck, Austria

[14] Department of Earth and Environmental Sciences, University of Illinois at Chicago, Chicago, IL, USA

[15] Department of Environmental Sciences, Rutgers University, New Brunswick, NJ, USA

[16] School of Environment, Earth and Ecosystem Sciences, The Open University, MK 7 6AA Milton Keynes, United Kingdom

[17] Department of Atmospheric & Oceanic Sciences, UCLA, USA

[18] Department of Earth & Environmental Sciences, Lamont Doherty Earth Observatory, Columbia University, Palisades, NY 10964, USA

[19] Department of Ecology and Evolutionary Biology, University of Arizona, USA

[20] Currently affiliated at: Center for Atmospheric and Environmental Chemistry, Aerodyne Research, Inc., USA

*Correspondence to*: Linda Kooijmans (linda.kooijmans@wur.nl)

**Abstract.** The uptake of carbonyl sulfide (COS) by terrestrial plants is linked to photosynthetic uptake of $CO_2$ by a shared diffusion pathway. Applying COS as a photosynthesis tracer in models requires an accurate representation of biosphere COS fluxes, but these models have not been extensively evaluated against field observations of COS fluxes. In this paper, the COS flux as simulated by the Simple Biosphere Model, version 4 (SiB4) is updated with the latest mechanistic insights and evaluated with site observations from different biomes: one evergreen needleleaf forest, two deciduous broadleaf forests, three grasslands, and two crop fields spread over Europe and North America. To account for the effect of atmospheric COS mole





fractions on COS biosphere uptake, we replaced the fixed COS mole fraction originally used in SiB4 with spatially and temporally varying COS mole fraction fields. The lower COS mole fractions in the late growing season reduces COS uptake

rates in agreement with observations. We also replaced the empirical soil COS uptake model in SiB4 with a mechanistic model that represents both uptake and production of COS in soils, which improves the match with observations over agricultural fields and fertilized grassland soils. SiB4 was capable of simulating the diurnal and seasonal variation of COS fluxes in the boreal, temperate and Mediterranean region. The daytime vegetation COS flux is on average 8 ± 27 % underestimated, albeit with large variability across sites. On a global scale, our model modifications caused a drop in the COS biosphere sink from

922 Gg S yr$^{-1}$ in the original SiB4 model to 753 Gg S yr$^{-1}$ in the updated version. The largest drop in fluxes was driven by lower atmospheric COS mole fractions over regions with high productivity, which highlights the importance of accounting for variations in atmospheric COS mole fractions. The change to a different soil model, on the other hand, had a relatively small effect on the global biosphere COS sink. The small role of the modeled soil component in the COS budget supports the use of COS as a global photosynthesis tracer.

**1 Introduction**

Carbonyl sulfide (COS) uptake by the terrestrial biosphere is the main sink of atmospheric COS (Whelan et al., 2018). COS uptake in plants is closely related to photosynthetic $CO_2$ uptake through its shared uptake pathway through plant stomata and, as a consequence, COS can be used to constrain the carbon and water cycles (Seibt et al., 2010; Stimler et al., 2010; Whelan et al., 2018). Key plant processes such as photosynthesis and transpiration are difficult to observe at scales larger than the leaf

level because they are contained within the net $CO_2$ flux and evapotranspiration and are not separable from other fluxes. Constraints on these fluxes are therefore needed to improve terrestrial biosphere models to better simulate the responses of photosynthesis and stomatal gas exchange to a changing climate. Recently, COS has been shown to be of added value for understanding changes in plant uptake, e.g., the inhibition of photosynthesis during a heat wave (Wohlfahrt et al. 2018), the growth of the terrestrial gross primary production (GPP) during the twentieth century (Campbell et al., 2017), and changes in

transpiration (Berkelhammer et al., 2020; Wehr et al., 2017). To further advance COS as a constraint on the carbon and water cycles in models requires an accurate representation and evaluation of COS biosphere fluxes in models.

Biosphere COS exchange has been implemented in land surface models such as the Simple Biosphere Model, version 3 (SiB3; Berry et al. 2013), the Organising Carbon and Hydrology In Dynamic Ecosystems model (ORCHIDEE; Launois et al. 2015a;

Maignan et al. 2020), the Lund-Potsdam-Jena model (LPJ) and the Community Land Model (CLM4) (Launois et al. 2015a). Estimates of the global biosphere uptake of COS from these models and other approaches range between 368 and 1845 Gg S yr$^{-1}$ with a mean of 1084 Gg S yr$^{-1}$ over 9 different studies as summarized in Table 1 (Kettle et al. 2002; Montzka et al. 2007; Suntharalingam et al. 2008; Berry et al. 2013; Launois et al. 2015a; Kuai et al. 2015; Wang et al. 2016; Maignan et al. 2020; Ma et al. 2021). These estimates were made through different approaches, such as scaling COS vegetation uptake to the net



(NPP) or gross primary production (GPP), and more recently also from mechanistic implementations (Table 1). The mechanistic implementations of COS vegetation uptake in the biosphere models yield a smaller range of 688-775 Gg S yr$^{-1}$ than when the COS vegetation uptake is scaled to the $CO_2$ vegetation sink (Table 1). The global soil COS sink estimates range from 130 to 510 Gg S yr$^{-1}$, but with most estimates between 130 and 176 Gg S yr$^{-1}$. However, land surface models have still not adopted the available mechanistic soil models from either Sun et al. (2015) or Ogée et al. (2016).


It was pointed out by Ma et al. (2021) that the temporal and spatial variability of atmospheric COS mole fractions has considerable influence on the COS biosphere uptake; e.g. a seasonal amplitude of ~100-200 parts per trillion (ppt) around an average of ~500 ppt affects the fluxes by ~20-40 %. In contrast to $CO_2$, where a seasonal amplitude of ~6-7 ppm around ~410 ppm affects the fluxes only by ~2 %. Although some of the previous studies considered the variable COS mole fraction already

(Berry et al., 2013, Kuai et al., 2015, Wang et al., 2016), it has not yet been adopted as a standard approach (Maignan et al. 2021, Ma et al., 2021).

Inverse modeling studies that account for all known sources and sinks of COS imply a missing source of COS in the tropical region (Berry et al. 2013; Le Kuai et al. 2015; Ma et al. 2021). Ma et al. (2021) revealed considerable seasonal variations of

the missing source. Yet, the exact reason for this missing source has not been resolved. Although the missing source can be anthropogenic or from the tropical ocean (Launois et al. 2015b; Kuai et al. 2015; Lennartz et al. 2017, 2019), an overestimated tropical biospheric sink cannot be ruled out. Moreover, Ma et al. (2021) identified a missing sink at the higher latitudes that required larger uptake in summer. This missing sink could be explained by an underestimated biosphere sink, and would be equivalent to a 6 % underestimation of the biosphere sink above 30 °N (Ma et al., 2021).


A source of uncertainty for COS uptake by land surface models is that simulations have not been extensively compared against field observations because field measurements of ecosystem and soil fluxes are sparse. Yet, several research groups have performed field observations of COS ecosystem fluxes in the last decade (Asaf et al. 2013; Maseyk et al. 2014; Commane et al. 2015; Kooijmans et al. 2017; Wehr et al. 2017; Yang et al. 2018; Spielmann et al. 2019; Berkelhammer et al. 2020; Vesala

et al., in prep) with observations covering evergreen needleleaf forests, deciduous broadleaf forests, grasslands, and crop fields. These experimental efforts now offer the possibility to compare model simulations of COS biosphere exchange against field observations from different biomes.

In this paper, we take advantage of these field measurements for comparison with the latest version of the SiB model, version

4 (SiB4) and to evaluate the calculated global COS biosphere flux. When compared to SiB3 (Berry et al. 2013), SiB4 has enhanced capabilities to simulate variable carbon pool allocation, prognostic phenology, land cover heterogeneity, and crop phenology (Haynes et al. 2019a). We evaluate seasonal and diurnal cycles of ecosystem COS fluxes and the representativeness of nighttime COS uptake, where the latter is important for an accurate COS sink estimate. We furthermore update the SiB4



model with knowledge obtained on soil exchange of COS during the last decade by implementing the mechanistic soil model

from Ogée et al. (2016) for COS soil uptake and production rates varying with biome after Meredith et al. (2018, 2019). Furthermore, we replace the fixed atmospheric COS mole fraction of 500 pmol mol$^{-1}$ with spatially and temporally varying COS mole fraction fields obtained from an inversion with the TM5-4DVAR atmospheric transport model (Ma et al., 2021). We diagnose possible biases from the model-observation comparison and conclude with recommendations for further improvement of the model.









**Table 1. Global vegetation, soil and biosphere COS sink estimates presented in literature and this study (Gg S yr-1)**

| | Kettle et al. (2002) | Montzka et al. (2007) | Suntharalingam et al. (2008) | Berry et al. (2013) | Kuai et al. (2015) | Wang et al. (2016) | Launois et al. (2015a) | | | Maignan et al. (2021) | Ma et al. (2021) | This study Original SiB4 | This study Modified SiB4 |
|---|---|---|---|---|---|---|---|---|---|---|---|---|---|
| Vegetation | 238 | 1115 | 490 | 738 | 775 | 688 | 1335 (708[e]) | 1069 (663[e]) | 930 (772[e]) | 756 | - | 776 | 664 |
| Method | NPP scaling | NPP scaling | NPP scaling | SiB3 | SiB3 | SiB3 | GPP scaling, ORC | GPP scaling, LPJ | GPP scaling, CLM4 | ORC | SiB4 | SiB4 | SiB4[d] |
| Soil | 130 | 127[a] | 130[a] | 355 | 176 | 159 | 513 (283[e]) | 513 (398[e]) | 513 (507[e]) | - | - | 146 | 89 |
| Method | Soil model[b] | Soil model[b] | Soil model[b] | SiB3[c] | SiB3[c] | SiB3[c] | Scaling to H₂ deposition | | | | SiB4[c] | SiB4[c] | SiB4[d,f] |
| Total bio-sphere sink | 368 | 1242 | 620 | 1093[d] | 951 | 847 | 1845 | 1579 | 1440 | | 1053 (851[d]) | 922 | 753 |

[a] Adopted from Kettle et al. (2002)

[b] Following Kesselmeier et al. (1999)

[c] Scaled to heterotrophic $CO_2$ respiration

[d] Considering a variable mixing ratio

[e] After optimization

[f] Following Ogée et al. (2016)



## 2. Methodology

### 2.1 SiB4 model

The Simple Biosphere Model version 4 (SiB4) is a mechanistic, prognostic land surface model that integrates heterogeneous land cover, environmentally responsive prognostic phenology, dynamic carbon allocation, and cascading carbon pools from live biomass to surface litter to soil organic matter (Haynes et al. 2019a,b). SiB4 predicts vegetation and soil moisture states, land surface energy and water budgets, and the terrestrial carbon cycle. Rather than relying on satellite data as in SiB3, SiB4 fully simulates the terrestrial carbon cycle by using the carbon fluxes to determine the above and belowground biomass, which in turn feeds back to impact carbon assimilation and respiration (Haynes et al. 2020). SiB4 predicts plant phenology, divided into different stages, allowing the change of photosynthetic activity over the season through specified maximum RuBisCO velocities in each phenological stage. To classify land surface vegetation, SiB4 uses plant functional types (PFTs), which group plants according to their function and physical, physiological, and phenological characteristics. In addition to nine natural vegetation PFTs, SiB4 includes three specific crops (maize, soybeans, and winter wheat), and two generic crops (C3 and C4) following the crop phenology model developed by Lokupitiya et al. (2009). SiB4 includes land cover heterogeneity by simulating multiple PFTs per grid cell.

### 2.1.1 COS plant and soil uptake after Berry et al. (2013)

COS plant uptake in the SiB4 model has been described by Berry et al. (2013) and is simulated as a series of conductances ($g_t$) from the leaf boundary layer to the site of COS hydrolysis in the mesophyll cells. These conductances include the conductance from canopy air to the leaf surface, or leaf boundary layer conductance ($g_b$), the stomatal conductance ($g_s$), and the internal conductance ($g_{cos}$). The latter represents both the diffusion of COS to the mesophyll cells and the efficiency of the leaf mesophyll carbonic anhydrase (CA) to hydrolyze COS. This leads to the following equation for the COS uptake rate by vegetation:

$$F_{cos,veg} = C_a \frac{1}{\frac{1.94}{g_s} + \frac{1.56}{g_b} + \frac{1}{g_{cos}}} = C_a g_t,$$

(1)

where $F_{cos,veg}$ is the COS vegetation uptake rate (pmol m$^{-2}$ s$^{-1}$) and $C_a$ is the COS mole fraction in the canopy air space (pmol mol$^{-1}$) calculated from the mixed layer COS mole fraction (standard 500 pmol mol$^{-1}$, but see Sect. 2.1.3.) taking into account uptake of COS by soil and vegetation in the previous timestep. $g_s$ and $g_b$ are the stomatal and boundary layer conductances to water vapor (mol m$^{-2}$ s$^{-1}$), respectively, and are scaled with diffusivity ratios to account for the different diffusivity rates of COS and H$_2$O (Seibt et al., 2010; Stimler et al., 2010). The stomatal conductance $g_s$ is derived following the Ball-Berry photosynthesis-conductance model as modified by Collatz et al. (1992) and $g_b$ follows the formulations described by Sellers et al. (1996). The internal conductance $g_{cos}$ is assumed to scale with maximum carboxylation rate of RuBisCO, $V_{max}$ (μmol m$^{-2}$ s$^{-1}$) (Berry et al. 2013), inspired by previous findings that both CA activity (Badger and Price, 1994), and mesophyll





conductance (Evans et al., 1994) scale with $V_{max}$ in C3 species. In SiB4, $V_{max}$ is adjusted to canopy temperature ($T_{can}$) following (Sellers et al., 1992):

$$V_{maxT} = V_{max} 2.1^{0.1(T_{can}-298.0)},$$
(2)

$g_{cos}$ is then described as:

$$\begin{cases} g_{cos} = \alpha \cdot V_{COS} \\ V_{cos} = V_{maxT} \cdot F_{LC} \cdot F_{RZ} \cdot \left(\dfrac{p}{p_{0sfc}}\right) \cdot \dfrac{T_{can}}{T_0} \end{cases}$$
(3)

where $F_{LC}$ is a factor scaling the flux from a single leaf to the canopy that considers the canopy profile of absorbed photosynthetically active radiation (Sellers et al., 1996), $F_{RZ}$ is the rootzone water potential, an empirical scaling factor that reduces the biochemical activity when little soil moisture is available (e.g. during extended periods of drought), $p/p_{0sfc}$ adjust the fluxes for altitude, where $p$ is atmospheric pressure (hPa) and $p_{0sfc}$ the reference surface pressure (1000 hPa), and $T_{can}/T_0$

scales the flux to a reference temperature at $T_0$ = 273.15 K. A calibration term $\alpha$ was added to scale $g_{cos}$ to COS flux observations of controlled gas exchange measurements (Stimler et al., 2010, 2011), which resulted in $\alpha$ = 1200 for C3 and 13000 for C4 species (Berry et al. 2013). These numbers were later updated to $\alpha$ = 1400 and 8862 for C3 and C4 species, respectively, after updates were made to the SiB model. Berry et al. (2013) already noted that the $\alpha$ value did not constrain the variability between plant species well, likely due to plant variability in CA activity and/or differences in mesophyll

conductance. In Sect. 2.3 we explain how we use field measurements to explore whether we can refine $\alpha$ values for different plant functional types separately and to make it variable over time.

The enzyme CA is expressed in microbial communities in soils as well, leading to COS uptake by soils (e.g. Kesselmeier et al., 1999; Meredith et al. 2019). In SiB4, COS uptake in soils (hereafter called "the Berry soil model") is coupled to

heterotrophic $CO_2$ respiration under the assumption that in more productive regions there would be more litter and surface soil carbon for respiration, and these richer carbon environments would have more CA as well (Yi et al., 2007). Additionally, COS soil uptake in the model is regulated by diffusion, controlled by soil porosity and the fraction of water filled pore space (Van Diest and Kesselmeier, 2008; Ogée et al., 2016; Sun et al., 2015; Whelan et al., 2016). Initial implementations of soil COS uptake made calculations for the entire soil column, but subsequent model versions considered only uptake in the top 20 cm

of the soil (Wang et al. 2016), thereby decreasing global soil uptake estimates from 355 (Berry et al. 2013) to 159 Gg S yr$^{-1}$ (Wang et al. 2016). In the next section, we describe our update to the SiB4 model based on advances in our knowledge on COS soil exchange obtained during the last decade.





### 2.1.2 Mechanistic COS soil model after Ogée et al. (2016)

Field and laboratory experiments in the last decade showed that COS is not only taken up by soil but is also produced due to
abiotic thermal degradation and photodegradation of soil organic matter and is especially enhanced in agricultural soils
(Maseyk et al. 2014; Whelan and Rhew 2015; Meredith et al. 2018; Kaisermann et al. 2018a). Besides COS soil production
being enhanced in fertilized soils, COS uptake was shown to be diminished in fertilized soils (Kaisermann et al. 2018b). These
effects of nutrient fertilization on soil COS exchange were initially not simulated in the SiB4 model.

New empirical soil models (Whelan et al., 2016) and mechanistic models (Ogée et al., 2016; Sun et al., 2015) were developed
during the last decade. The mechanistic models describe the uptake and production pathways together with COS diffusion in
a soil column. Ogée et al. (2016) derived a simplified analytical solution assuming a soil column with uniform temperature,
soil moisture, and porosity and steady state conditions for comparison against laboratory measurements. The model from Ogée
et al. (2016), hereafter called "the Ogée soil model", was then used by several laboratory studies to study patterns in uptake
and production of COS in soils (Meredith et al. 2018; 2019; Kaisermann et al. 2018a,b). Due to these efforts, there are now
reaction rate parameter values available for a range of biomes and land use types. Because these reaction rate values were
derived by fitting the Ogée soil model on data from mesocosm experiments, they should be used in combination with this
model to estimate ecosystem-scale soil COS fluxes. Also, compared to the COS soil model proposed by Sun et al. (2015), the
steady state solution of the Ogée soil model is computationally inexpensive and therefore more suitable for implementation in
SiB4 for global COS soil flux calculations. In the following paragraphs we describe the implementation of the Ogée soil model
in SiB4.

For field conditions (assuming a zero COS vertical gradient at the bottom of the soil layer and steady state) the COS soil flux
(mol m$^{-2}$ s$^{-1}$) calculation simplifies to (Ogée et al., 2016):

$$F_{COS,soil} = \sqrt{kB\theta D} \cdot \left( C_a - \frac{z_1^2 P}{D} \left( 1 - \exp\left(\frac{z_p}{z_1}\right) \right) \right), \quad\quad (4)$$

where $k$ is the CA reaction rate (s$^{-1}$), $B$ (m$^3$ water m$^{-3}$ air) the solubility of COS in water that relates to Henry's law constant
and depends on temperature, $\theta$ the soil water content (m$^3$ m$^{-3}$), $D$ the soil COS diffusivity (m$^3$ air m$^{-1}$ soil s$^{-1}$), $C_a$ the COS mole
fraction at the soil-air interface, $z_1^2 = D/(kB\theta)$, and $P$ the COS production rate (mol m$^{-3}$ s$^{-1}$) uniform over depth $z_p$ (here assumed
to be 1.0 m). For details of the model calculations we refer to Ogée et al. (2016), here we provide the information specific for
the implementation in SiB4. We assume $C_a$ to be identical to the COS mole fraction in the canopy air space. While
implementing and testing the model we recognized the strong dependence of the soil fluxes on soil porosity, choice of tortuosity
functions, and the SiB4 soil layer selected for temperature and soil moisture. For the calculation of $D$ we used the SiB4 soil
porosity (m$^3$ m$^{-3}$; calculated from sand fractions following Lawrence and Slater (2008)) that accounts for the volume of ice in
the soil. The simulated soil water content and soil temperature are taken from the top 5 cm soil layer, where most of the COS



uptake takes place. *D* also depends on tortuosity functions that describe the tortuous movement through the air- or water-filled
pore space. Several tortuosity functions are described in the literature and also Ogée et al. (2016) acknowledged that the response of the soil COS fluxes to soil moisture varied with the chosen tortuosity functions. We chose the tortuosity functions of Deepagoda et al. (2011) for air and Millington and Quirk (1961) for water, as these functions do not require a pore-size distribution parameter, which facilitates its implementation in SiB4.

COS is taken up in soils through hydrolysis in soil water, where the main consumption is enzymatic, and thus depending on soil CA enzyme activity. Here, and following other studies (i.e. Ogée et al. 2016, Meredith et al. 2019), we expressed the CA reaction rate k relative to the uncatalyzed reaction rate ($k_{uncat}$) at a reference temperature ($T_{ref}$) and pH:

$$k = f_{CA} k_{uncat} \frac{x_{CA}(T)}{x_{CA}(T_{ref})}, \tag{5}$$

where $f_{CA}$ is the CA enhancement factor, $k_{uncat}$ varies with soil pH according to Elliott et al. (1989) and $x_{CA}(T)$ and $x_{CA}(T_{ref})$ are temperature response functions (Ogée et al., 2016).


Meredith et al. (2019) collected soils from 20 sites from different biomes. Using controlled laboratory measurements, they derived $k_{cat}$, $k_{uncat}$ and $f_{CA}$ from a range of biomes and land use types. In SiB4 we used the biome averaged $f_{CA}$ from Meredith et al. (2019) for calculation of COS soil uptake across different PFTs (Table 2).

The COS production was defined by Ogée et al. (2016) as a temperature response function modulated by the soil redox potential. Meredith et al. (2018) also measured COS production at a temperature range between 10 and 40 °C for a range of biomes. Measurements were then fitted to an exponential model:

$$P = a \exp(bT_{soil}). \tag{6}$$

We used this exponential temperature model and the biome averaged *a* and *b* (Table 2) in our calculation of *P* in SiB4. We assume here that the controlled laboratory measurements by Meredith et al. (2018; 2019) can be used to estimate soil fluxes
under field conditions.






**Table 2. Biome-averaged uptake and production parameters after Meredith et al. (2018; 2019).**

|  | Production parameters[a] | | Uptake parameter[b] |
|---|---|---|---|
|  | $a \pm$ std (pmol m$^{-3}$ s$^{-1}$) | $b \pm$ std (1/°C) | $f_{CA}$ |
| Grass | 2.20 ± 0.5 | 0.096 ± 0.005 | 45000[c] |
| Evergreen forest | 4.86 ± 2.7 | 0.101 ± 0.015 | 32000[d] |
| Deciduous forest | 4.94 ± 0.7 | 0.107 ± 0.002 | 32000[d] |
| Agriculture | 9.59 ± 7.3 | 0.104 ± 0.004 | 6500 |
| Desert/Bare soil | 5.60 ± 5.1 | 0.050 ± 0.010 | 13000[e] |

[a]Based on Meredith et al. (2018)
[b]Based on Meredith et al. (2019)
[c]Measurements represent tropical grassland
[d]Measurements represent temperate coniferous and temperate broadleaf forests
[e]Measurements represent desert soil

### 2.1.3 Variable atmospheric COS mole fractions

The atmospheric COS mole fraction in the planetary boundary layer affects both the COS vegetation and soil flux calculations (Eq. (1) and (4)). In SiB4 a standard constant "place-holder" COS mole fraction of 500 pmol mol$^{-1}$ is used. Ma et al. (2021) estimated that the global biosphere sink would decrease from 1053 Gg S yr$^{-1}$ to 851 Gg S yr$^{-1}$ if the fixed COS mole fraction were replaced with monthly mean fields that account for the drawdown of COS near the surface in the peak growing season. We thus changed the prescribed COS mole fraction from a fixed value to one varying in space and time, including seasonal and diurnal variability. To this end we used the surface COS mole fraction fields at a global resolution of 4° × 6° (latitude × longitude) at 3-hourly time steps as retrieved from an atmospheric transport inversion performed with TM5-4DVAR by Ma et al. (2021) using the chemistry transport model TM5 in which COS exchange was recently implemented. Atmospheric measurements of COS mole fractions at 14 sites from the National Oceanic and Atmospheric Administration (NOAA) flask network (Montzka et al., 2007) were used to optimize the sources and sinks of COS. Here, we used global 2D surface layer fields of COS mole fractions that were determined for the period 2016-2018 and repeated the average over those years as input to the SiB4 mixed layer COS mole fraction for each year in the simulation (see global maps of monthly mean surface COS mole fractions in supplementary figure S13 of Ma et al. (2021)). The changing (e.g. lower) COS mole fractions would lead to lower COS uptake rates, but would in turn also lead to a smaller drop in COS mole fractions, this feedback is currently not accounted for.

### 2.1.4 Simulations

We used meteorological data from the Modern-Era Retrospective Analysis for Research and Applications, version 2 (MERRA), which are available from 1980 onwards (Gelaro et al., 2017) as meteorological forcing to SiB4. To ensure realistic

rainfall, the convective and large-scale precipitation values were scaled such that the monthly total rainfall matches with the monthly precipitation in the Global Precipitation Climatology Project, Version 1.2 product (Huffman et al. 2001; Baker et al.

2010; Haynes et al. 2019a,b). Up to 10 PFTs per grid cell (at 0.5° × 0.5° resolution) are prescribed following PFT maps based on MODIS data (Lawrence and Chase, 2007). The soil characteristics such as sand fraction (used for the calculation of soil porosity) are provided by the International Geosphere-Biosphere Programme (IGBP) Global Soil Data Task Group (2000).

We run SiB4 from 2000 to 2020, the simulations were preceded by a spinup iterating five times over the years 2000-2020 to

initialize the carbon pools to reach steady state. $CO_2$ mole fractions were held constant at 370 µmol mol$^{-1}$ during spinup and simulations. We performed two sets of four simulations (global and site level) with the same driver data and settings, but with a different temporal resolution of the output: 1. For global simulations, we used monthly averaged output. Moreover, SiB4 simulates multiple PFTs per grid cell. These were averaged, weighted by the fraction of land area occupied by each PFT; 2. To compare SiB4 with site observations (listed in Table 3), we run the SiB4 model with 3-hourly output for only the grid cells

(at 0.5° x 0.5° resolution) in which the sites are located. For comparison with observations we selected the PFT that best represents the measurement site.

We run SiB4 with four different configurations:

1) the original SiB4 model containing the standard COS mole fraction of 500 pmol mol$^{-1}$ and the Berry soil model

(SiB4_500_Berry);

2) the Ogée soil model and the standard COS mole fraction of 500 pmol mol$^{-1}$ (SiB4_500_Ogee);

3) the Berry soil model and variable COS mole fractions (SiB4_var_Berry), and

4) the Ogée soil model and variable COS mole fractions (SiB4_var_Ogee).

**2.2 Field observations**

We use existing field observations for comparison with the SiB4 model simulations. Several studies have collected field data in the last two decades and we used those sites where continuous hourly measurements of ecosystem COS fluxes are available for at least a month. The site locations, some of their characteristics, and basic information on the observations are summarized in Table 3. The locations of the sites are indicated in Fig. 1. The measurements represent evergreen needleleaf forest (ENF), deciduous broadleaf forest (DBF), maize (MAI), winter wheat (WWT), and C3 grasslands (C3-GRA), more specifically alpine

grassland, prairie grassland, and savannah grassland.





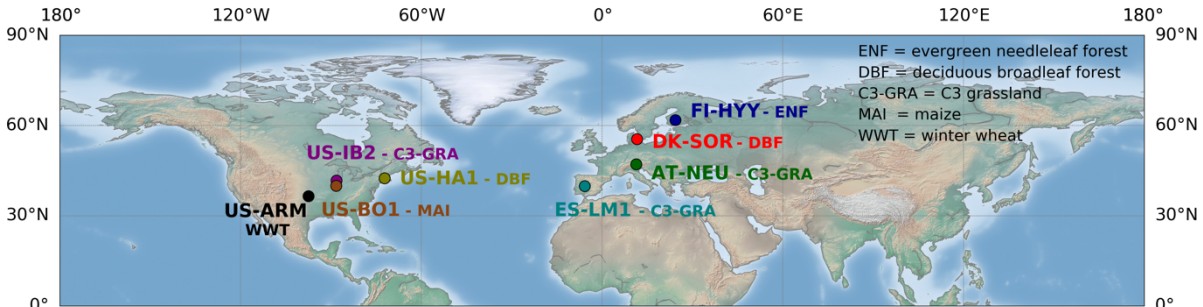

**Figure 1. Location of measurement sites and information on the PFT they represent.**

All COS observations were made at FLUXNET, ICOS or AmeriFlux sites with the benefit that additional long-term measurements of $CO_2$ and water exchange (Pastorello et al., 2020) are often available (see Table S1 for an overview), allowing the evaluation of the SiB4 phenology when COS flux observations do not extend to a full growing season. Most of the ecosystem observations were made using the eddy-covariance (EC) technique. Kohonen et al. (2020) summarized the different EC processing steps used by the different studies. Only at US-IB2, US-BO1, and for a part of the dataset at US-HA1 (in 2011),

the ecosystem fluxes are derived by COS concentration gradients using the flux-profile (FP) technique (Berkelhammer et al., 2020; Commane et al., 2015). Ecosystem fluxes are corrected for storage of COS in the canopy airspace using collocated canopy COS profile measurements when available (FI-HYY and US-HA1).

Most of the selected sites have *in situ* COS soil flux observations available for at least a part of the total measurement period

so that the COS uptake by vegetation can be derived from observed ecosystem fluxes. Measurements were collected using soil chambers, except at US-HA1, where atmospheric profile measurements near the surface were used to calculate the soil fluxes in 2012 and 2013.

Measurement datasets also include COS mole fractions above the canopy (except for US-ARM). These measurements have

been calibrated against the NOAA-2004 COS calibration scale. Only at US-HA1 the COS mole fractions are not calibrated (Commane et al., 2015), but validated against COS flask measurements at the station, which are part of the NOAA flask measurement network (Montzka et al., 2007).

For further details about the site characteristics and measurement and processing procedures we refer to the original data

publications as reported in Table 3 and Table S1.

For evaluation of the model against observations we calculate the mean bias error (MBE; pmol $m^{-2}$ $s^{-1}$) and root mean square error (RMSE; pmol $m^{-2}$ $s^{-1}$) for monthly, daytime, and nighttime average fluxes.





**Table 3.** Site and measurement information of field observations that are used for comparison with the SiB4 model. Sites are shown
from high to low latitude. PFTs covered by the sites are evergreen needleleaf forest (ENF), deciduous broadleaf forest (DBF), C3
grassland (C3-GRA), maize (MAI), and winter wheat (WWT). The ecosystem and soil flux measurement techniques are indicated
as eddy-covariance (EC), flux-profile (FP), chamber measurements, or are not available (NA). Mean annual temperature and mean
annual precipitation are shown in Table S1.

|  | Lat (°N), Lon (°E) | SiB4 PFT | Year | Months | Soil flux | Ecosystem flux | Reference |
|---|---|---|---|---|---|---|---|
| Hyytiälä, Finland (FI-HYY) | 61.8, 24.3 | ENF | '13 - '17 | Jan-Nov | Chamber | EC | Kooijmans et al. 2017; 2019; Sun et al. 2018; Vesala et al. in prep. |
| Sorø, Denmark (DK-SOR) | 55.5, 11.6 | DBF | '16 | Jun | Chamber | EC | Spielmann et al. 2019 |
| Neustift, Austria (AT-NEU) | 47.1, 11.3 | C3-GRA (alpine grassland) | '15 | Jun-Aug | Chamber | EC | Spielmann et al. 2019; 2020 |
| Harvard Forest, US (US-HA1) | 42.5, -72.2 | DBF (mixed forest dominated by DBF) | '11 - '13 | Apr-Oct | FP ('12-'13) | FP ('11) + EC ('12-'13) | Commane et al. 2015; Wehr et al. 2017 |
| Fermilab, US (US-IB2) | 41.8, -88.2 | C3-GRA (prairie grassland) | '16 - '17 | Apr-Oct | NA | FP | Berkelhammer et al. 2020 |
| Bondville, US (US-BO1) | 40.0, -88.3 | MAI | '15 | Jul/Sep | NA | FP | Berkelhammer et al. 2020 |
| Majadas, Spain (ES-LM1) | 39.9, -5.8 | C3-GRA (savannah grassland) | '16 | May | Chamber | EC | Spielmann et al. 2019 |
| ARM Southern Great Plains, US (US-ARM) | 36.6, -97.5 | WWT | '12 | Apr-May | Chamber | EC | Maseyk et al. 2014 |

### 2.3. Calibration factor α

Berry et al. (2013) used the calibration factor α to scale $g_{cos}$ to match the simulated COS vegetation flux with laboratory
measurements. They noted that the α value did not constrain the variability between plant species well, likely due to plant
variability in CA activity and/or mesophyll conductance. Here, we derived $α_{obs}$ from COS field measurements. This analysis
is meant to explore its variability over time and the necessity to define α values specific for different PFTs. We did not retain
$α_{obs}$ for global simulations.


We derived $g_t$ from measurements of canopy vegetation fluxes ($F_{COS,veg}$ = ecosystem – soil fluxes) and simulated COS mole
fractions in the canopy airspace $C_a$:



$$g_{t,obs} = \frac{F_{COS,veg}}{C_a}. \tag{9}$$

Then, rewriting Eq. (1) and (3) and adopting $g_s$, $g_b$ and $g_{cos}$ from SiB4 site simulations we calculated $\alpha_{obs}$ as:

$$\alpha_{obs} = \frac{-1}{\frac{V_{cos}}{g_{t,obs}}\left(g_{t,obs}\frac{1.94}{g_s} + g_{t,obs}\frac{1.56}{g_b} - 1\right)}. \tag{10}$$

$\alpha_{obs}$ was calculated for daytime hours (10 – 15 hr local time) in periods with photosynthetically active vegetation, which

excludes data points of FI-HYY when plants are dormant in winter (November to April), and after the simulation of harvest at US-ARM. Raw $\alpha_{obs}$ data points were considered an outlier when their value extends 1.5 times the 25-75 percentile range outside the quartiles and were removed from the analysis.

This analysis requires that field measurements of ecosystem and soil fluxes are available. Under the assumption that both $V_{max}$ (and thus photosynthesis) and the soil flux are accurately simulated, the application of $\alpha_{obs}$ would result in simulated COS

vegetation fluxes that match with observations.

## 3. Results and discussion

First, we evaluate the SiB4 COS flux against observations (Sect. 3.1). The accuracy of the ecosystem flux is controlled by several factors, such as the accuracy in leaf phenology, differences in accuracy of the daytime and nighttime COS vegetation flux, the accuracy of the soil flux (of both the Berry and Ogée soil model), and the sensitivity to atmospheric COS mole

fractions. We discuss the role of each of these factors in the evaluation of SiB4 biosphere fluxes against observations. All results are based on the standard α values of 1400 and 8862 for C3 and C4 species, respectively. We present COS fluxes relative to the atmosphere (i.e. negative values indicate uptake by the ecosystem). Next, we study the variability of $\alpha_{obs}$ between different PFTs and across seasons (Sect. 3.2), to investigate remaining model-data mismatches in the COS vegetation flux that could potentially be solved by re-calibrating α. Finally, we present global estimates of the COS biospheric sink with different

model configurations (Sect. 3.3).

### 3.1 SiB4 COS flux evaluation and sensitivity

### 3.1.1 Seasonal variability

Simulated COS ecosystem fluxes capture the seasonal variation of monthly-averaged observations (Fig. 2), with similar results for vegetation fluxes alone (Fig. S1). Specifically, COS uptake peaked in summer in the simulations, as was observed at the

three sites that contain COS flux measurements across different seasons (Fig. 2a, d, e). At the other sites, COS ecosystem fluxes were only measured during one part of the growing season. Therefore, we also used multi-year NEE, GPP, and latent heat flux (LE) from FLUXNET, ICOS, and AmeriFlux to evaluate the SiB4 seasonality (Fig S2-4).



Based on the NEE, GPP, and LE observations (Fig. S2-4) the start and end of the growing season are typically well captured

by the SiB4 simulations. The timing and length of the growing season for grassland sites has been previously evaluated by Haynes et al. (2019b) using remotely-sensed leaf area index and showed that SiB4 was capable of simulating growing season timing and variability across temperature and precipitation gradients. Also, the timing of maximum NEE and GPP, which differs by PFT and climatic regions, was well captured; e.g., simulated and observed NEE and GPP peak in spring at the savannah grassland site ES-LM1 and at the winter wheat site US-ARM. All other sites show an observed and simulated summer

maximum carbon uptake. Only AT-NEU is an exception, with SiB4 predicting the peak net $CO_2$ uptake too late into the summer compared to the observations, which can be explained by grass cutting that was not included in SiB4. Crop harvesting was included in SiB4, but the exact timing was difficult to simulate due to local weather events and considerations other than crop ripening. For example, at the US-ARM site the winter wheat harvest was on average simulated at DOY 136 for the years 2000-2019, close to the actual moment of harvest in 2012: DOY 145. However, for 2012 (the year matching with COS flux

observations), the model simulates harvest almost 4 weeks earlier (DOY 118) than was actually the case, possibly because in 2012 the meteorological forcing data prescribed generally higher daytime temperatures than observed (the slope between observed and model air temperature was 1.14 in 2012), while in other years the model temperatures were similar to observations (the slope was on average $1.03 \pm 0.04$).

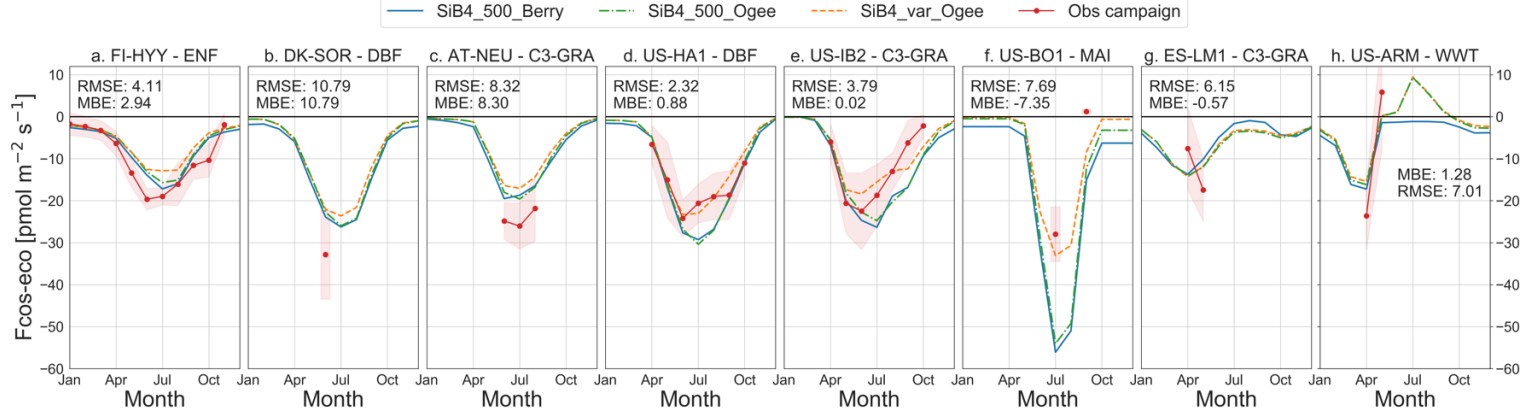

**Figure 2. Comparison of ecosystem COS flux seasonal cycles of observations (red) with different SiB4 model runs: the original SiB4 model with 500 pmol mol$^{-1}$ COS and the original Berry soil model (SiB4_500_Berry, blue, solid); a run with variable COS mole fractions and the Ogée soil model (SiB4_var_Ogee, orange, dashed); a run with 500 pmol mol$^{-1}$ COS and the Ogée soil model (SiB4_500_Ogee, green, dot-dash). Monthly averages are shown with the 1σ spread around the mean of observations. Negative values indicate uptake of COS by the ecosystem while positive values indicate COS emissions. The model simulations are from the**

**same year(s) in which observations were made. The MBE and RMSE (pmol m$^{-2}$ s$^{-1}$) are given for monthly average fluxes of the SiB4_var_Ogee run. Sites are presented from high to low latitude.**





For the sites where COS fluxes were only measured in one part of the growing season we assume that the timing of seasonal patterns in COS assimilation were well captured since seasonal patterns in NEE, GPP, and LE are properly simulated (Fig. S2-

4) and the model scales the CA activity with $V_{max}$, and $g_s$ with GPP.

We generally found larger underestimations of the ecosystem COS exchange at the higher latitudes (FI-HYY, DK-SOR, AT-NEU, Fig. 2a-c), which is consistent with findings by Ma et al. (2021) who found a missing sink at the higher latitudes that required larger uptake in summer (their Fig. 5b). The model-observation biases that we see in the ecosystem COS fluxes are

consistent with biases in GPP for some sites. For example, the underestimation of the COS ecosystem flux at DK-SOR, AT-NEU, and FI-HYY is consistent with underestimations of GPP (Fig. S3a-c), which will be further discussed in Sect. 3.1.3.

### 3.1.2 Effects of varying atmospheric COS mole fractions

Modifying the COS mole fractions to vary spatially and temporally significantly improved the comparison with observations in North America, as seen from the orange (variable COS) and green (fixed COS) line in Fig. 2d-f. Generally, COS mole

fractions are lower in the second half of the growing season, leading to lower COS uptake in that period. When a variable COS mole fraction was used, the MBE value in July-August improved from 9.0 to 2.0 pmol m$^{-2}$ s$^{-1}$ at US-HA1, from 7.2 to -0.9 pmol m$^{-2}$ s$^{-1}$ for US-IB2, and from 28.6 to 5.4 pmol m$^{-2}$ s$^{-1}$ in US-BO1. The influence of the COS mole fraction on the biosphere flux was largest at sites within or close to the Corn Belt in Midwestern US with strong biosphere COS uptake (see also Fig. 6) that therefore has the largest summer-time drop in COS mole fractions (Fig. S5d,e) or the lowest COS mole fraction in general

(Fig. S5f). The large COS uptake by maize is confirmed by the observed COS fluxes reaching ~70 pmol m$^{-2}$ s$^{-1}$ at midday (Fig. S6). In this region, the lower COS mole fractions lead to lower COS uptake, but would in turn lead to a smaller drop in COS mole fractions. As COS uptake and COS mole fractions are interconnected, SiB4 should ideally be directly coupled to an atmospheric transport model.

At other sites (Europe) the variable COS mole fractions did not improve the model-observation bias, but instead caused a slightly larger underestimation by the model. The comparison of COS mole fractions from the TM5-4DVAR inversion against those observed at the measurement sites (Fig. S5), did not indicate that the COS mole fractions were consistently better simulated over North America than over Europe. These results indicate that the underestimation of COS fluxes over Europe is not likely caused by an underestimation of the COS mole fractions.

### 3.1.3 Diurnal cycles

The monthly average ecosystem COS fluxes (Fig. 2) included both day- and nighttime fluxes, and soil and vegetation fluxes, which may each have its own biases. Figure 3 shows model-observation differences of vegetation COS uptake separated by day- and nighttime, defined as 10 − 15 hr and 21 − 03 hr local time, respectively. These day- and nighttime definitions exclude transitions between day and night (see diurnal cycles in Fig. S6 and S7). On average across all stations, simulated daytime



uptake between April through October was $1.9 \pm 6.5$ pmol m$^{-2}$ s$^{-1}$ (= $8 \pm 27$ %) lower than the observations. Even though the average model-observation difference is small, there is substantial variability between sites. The underestimation of daytime COS uptake of 19.3 pmol m$^{-2}$ s$^{-1}$ (34 %) at DK-SOR was exceptionally large, consistent with the underestimation of daytime GPP in the same period (13.1 μmol m$^{-2}$ s$^{-1}$, 34 %). The COS measurements at DK-SOR were made in June 2016, a period that was warmer than average at this site. As a result, observed GPP was 25 % higher in June 2016 compared to the 1996-2018

average (Fig. S3b). However, SiB4 simulates only a 7 % higher GPP in June 2016 compared to the 1996-2018 average. At the same time, LE is overestimated (Fig. S4b). These results point to an underestimation of the RuBisCo and CA enzyme activity, and thus $g_{cos}$, rather than $g_s$, as LE is not underestimated but even overestimated. Also at AT-NEU and FI-HYY the underestimation of COS vegetation uptake was consistent with underestimations of simulated GPP against longterm timeseries (Fig. S3a,c), with a 9 % underestimation of the COS vegetation flux and 13 % underestimation of GPP at FI-HYY in the

months June to August. At US-ARM we saw a switch from an underestimation to overestimation of daytime vegetation COS uptake over the months April and May, which may be due to COS emissions from other components than the soil, possibly associated with senescing vegetation, which is currently not represented in SiB4 (nor in other models). Overall, we found large variability in model-observation biases between sites, but no clear distinctions emerge from different PFTs for daytime fluxes.

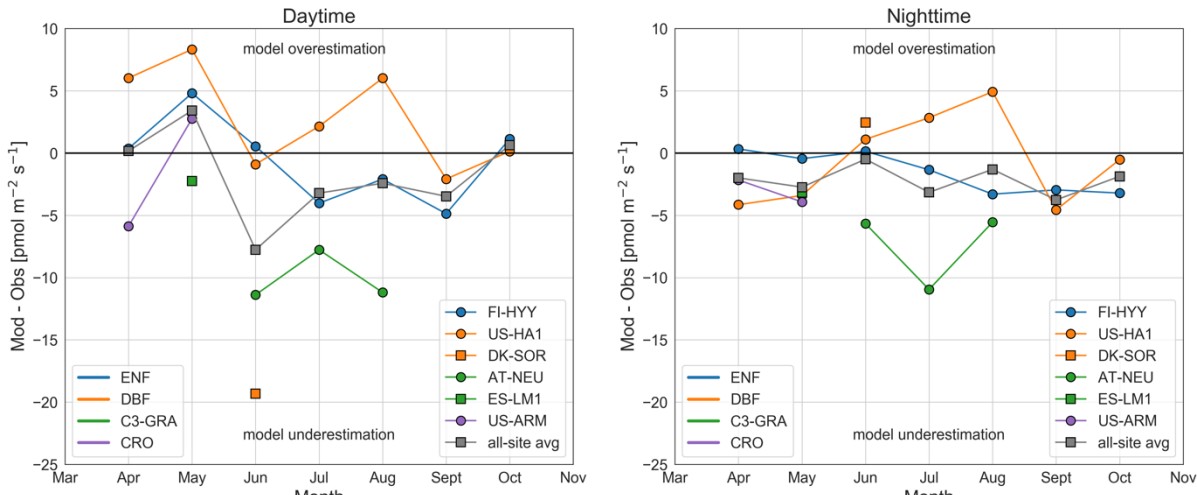


**Figure 3. Difference between model simulations and observations of monthly average COS vegetation fluxes (ecosystem – soil) for daytime data (10 – 15 hr local time; left) and nighttime data (21 – 03 hr local time; right). As ecosystem and soil fluxes are needed to obtain the vegetation flux, only sites with these data available are shown here. The model simulations were made with a variable**

**COS mole fraction and the Ogée soil model (SiB4_var_Ogee). Data are colored by PFT.**

The simulated nighttime uptake was on average $2.1 \pm 3.4$ pmol m$^{-2}$ s$^{-1}$ (= $35 \pm 57$ %) too small. Observed nighttime uptake was on average 25 % of the daytime uptake across sites between May-September, with the largest uptake at AT-NEU (11.0 pmol m$^{-2}$ s$^{-1}$), ES-LM1 (6.9 pmol m$^{-2}$ s$^{-1}$), and FI-HYY (5.9 pmol m$^{-2}$ s$^{-1}$). The small flux values during nighttime make the




model-observation comparison sensitive to the different correction and processing procedures that were used for the different datasets. Ecosystem fluxes were only storage-corrected for FI-HYY and US-HA1. Kooijmans et al. (2017) showed for FI-HYY that nighttime storage fluxes were on average ~1 pmol m$^{-2}$ s$^{-1}$ in summer. Additionally, some datasets are filtered based on a friction velocity threshold, while others are not. Kooijmans et al. (2017) noted that filtering data based on the friction velocity might bias the data to higher nighttime COS uptake as the uptake can be expected to be limited by the COS gradient

at the leaf boundary layer under low turbulence conditions. Given these differences between datasets, and the typically large random noise of COS flux measurements, the average underestimation may not be significant overall. Still, we found a substantial underestimation of the nighttime COS uptake at the C3-GRA sites AT-NEU and ES-LM1, and an overestimation in summer at the DBF sites US-HA1 and DK-SOR. These biases might point to inaccurate minimum stomatal conductance ($g_0$) values in SiB4, which are currently set to 10 mmol m$^{-2}$ s$^{-1}$ for all PFTs, except crop types (40 mmol m$^{-2}$ s$^{-1}$). Observed $g_0$

values at AT-NEU (10-65 mmol m$^{-2}$ s$^{-1}$, Wohlfahrt, 2004) are mostly higher than the 10 mmol m$^{-2}$ s$^{-1}$ used in SiB4 and support the hypothesis that the SiB4 $g_0$ is too low for this site. Similarly, estimates of $g_0$ at US-HA1 (3.1 mmol m$^{-2}$ s$^{-1}$, Wehr et al. 2017) point to a smaller value than used in SiB4 and could explain part of the overestimation of nighttime COS uptake at this site. These examples show that observations could help to obtain $g_0$ values for SiB4. Lombardozzi et al. (2017) made a literature overview of reported $g_0$ values per PFT and showed that $g_0$ was typically several times larger than the value of 10 mmol m$^{-2}$ s$^{-1}$

currently used in SiB4. We adopted the minimum values of Lombardozzi et al. (2017) in SiB4 to test the effect of a modified $g_0$ setting on the nighttime COS vegetation flux (see Table S2 and Fig. S8). Using these updated minimum values, the simulated nighttime COS uptake for C3-GRA improved at AT-NEU, but had larger biases for other sites and PFTs, especially DBF (Fig S8). As the $g_0$ values from Lombardozzi et al. (2017) did not consistently improve the nighttime COS uptake we did not adopt these as standard SiB4 settings.

**3.1.4 Soil fluxes**

The original SiB4 soil model scaled COS soil fluxes to heterotrophic $CO_2$ respiration, leading to COS uptake rates peaking at high temperatures in summer (Fig. 4, Fig. S9) and in conditions with sufficient soil moisture (Fig. 4g, Fig. S10g). The Ogée soil model also simulated COS uptake peaking at high temperatures, and lower uptake rates in winter compared to the Berry soil model (Fig. 4). In general, the COS uptake simulated by the Berry soil model matched well with observations at forest

sites (Fig. 4a, b, d), possibly because their approach was following a study on forest soils (Yi et al. 2007). The Ogée soil model underestimated the COS uptake at FI-HYY (Fig. 4a), but was closer to observations at the other forest sites US-HA1 and DK-SOR (Fig. 4b, d). The observed high soil COS uptake in April at FI-HYY is possibly related to snow melt and thawing of the soil and neither model captures this effect on soil COS exchange.





Soil COS emissions were observed at ES-LM1 and US-ARM. US-ARM was an agricultural site where emissions may build

up after the peak growing season in the period associated with senescence and harvest (Maseyk et al., 2014). The Berry soil

model did not simulate soil COS emissions (Fig. 4h). In contrast, the increase in COS emissions at the agricultural site US-

ARM was simulated by the Ogée soil model, although the increase of the emissions started later than in the observations. The

soil emissions of COS were not simulated at the C3-GRA site ES-LM1. However, the soil at ES-LM1 was fertilized (Weiner

et al. 2018), as well as that AT-NEU (Spielmann et al. 2020), which make these sites more representative of agricultural soils

rather than grassland soils. When ES-LM1 was simulated as an agricultural soil (the same code, but with different uptake and

production parameter values, see Table 2), the model showed COS emissions more consistent with observations (green line in

Fig. 4g). Also, the simulated fluxes at the AT-NEU site became smaller and in better agreement with observations when the

site was considered as an agricultural soil.

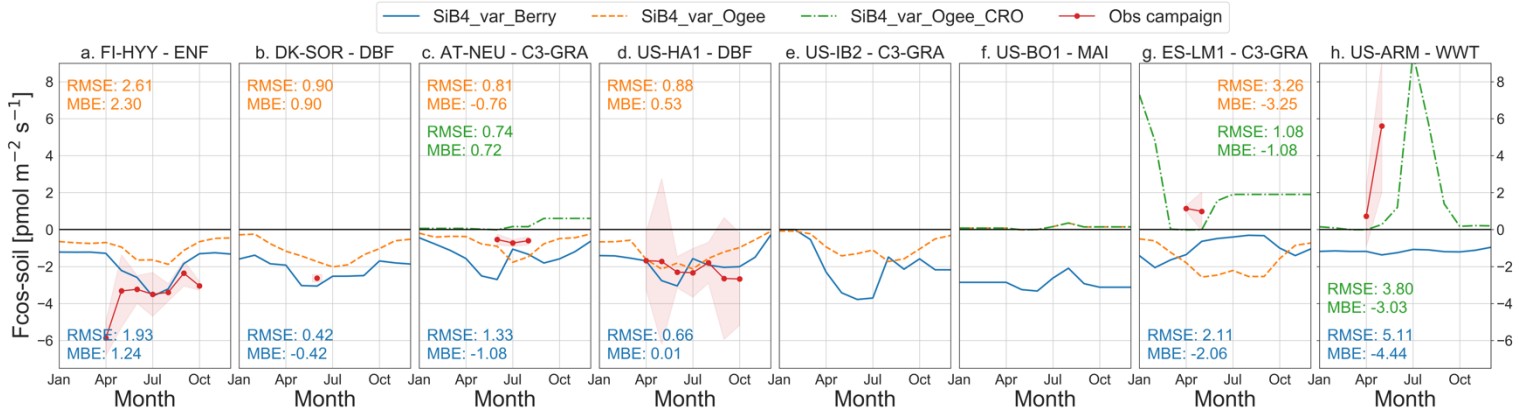


**Figure 4. Comparison of COS soil flux seasonal cycles of observations (red) with different SiB4 model runs: SiB4_var_Berry (blue, solid); SiB4_var_Ogee with the simulation representing the PFT type as indicated in the plot titles (orange, dashed); SiB4_var_Ogee_CRO with the simulation representing agricultural soil (Table 2) for sites AT-NEU, US-BO1, ES-LM1, US-ARM (green, dot-dash). No *in situ* observations of soil COS fluxes are available for US-IB2 and US-BO1. Monthly averages are shown**
**with the 1σ spread around the mean for observations. The model simulations are from the same year(s) in which observations were made. Negative values indicate uptake of COS by the ecosystem while positive values indicate COS emissions. The MBE and RMSE (pmol m⁻² s⁻¹) are given for monthly average fluxes for all model runs in their respective color. Sites are presented from high to low latitude.**

The accuracy of simulations of soil COS emissions depends on the accuracy of the production parameter $a$. The standard

deviation of the production parameter $a$ (7.3) is relatively large for agricultural soils compared to other soil types (Table 2)

and is an indication of the uncertainty of using a single production value in the SiB4 model. Reasons for this uncertainty can

be the local variability in soil characteristics like nitrogen content, which has been shown to correlate well with COS production

rates (Kaisermann et al. 2018b). Moreover, soil moisture and soil temperature were important parameters in the calculation of

the COS soil flux. In general, we found that the variability and absolute values of soil moisture and especially soil temperature

were well captured by the SiB4 model. We found a MBE across all sites of 0.01 m³ m⁻³ and 0.1 °C (RMSE 0.06 m³ m⁻³ and 2.1

°C) for soil moisture and temperature, respectively, calculated over all years available from the FLUXNET, AmeriFlux or





ICOS networks. Also, Smith et al. (2020) showed that SiB4 was capable of reproducing the drop in soil moisture as a result of a regional drought in Europe, albeit with a delay. We did not find consistent patterns in model-observation biases of the soil COS fluxes that were consistent with that of soil moisture or temperature (Fig. S9, S10). Still, the soil moisture observations

at US-ARM show a sharper drop in spring than the simulations (Fig. S10h), which could explain why the simulations show a delayed onset of soil COS emissions. Moreover, the exact role of thermal and photo-production of COS remains uncertain, as well as the interaction with soil organic matter and litter, and thereby limits the the accuracy of soil COS production simulations (Maseyk et al. 2014; Whelan and Rhew 2015; Meredith et al. 2018; Kaisermann et al. 2018a).

Overall, changing from the Berry soil model to the Ogée soil model had a relatively small effect on monthly average ecosystem fluxes (see SiB4_500_Berry (blue) and SiB4_500_Ogee (green) in Fig. 2), except for agricultural sites, where the Berry soil model lacked COS soil emissions that contribute to fluxes at those sites.

### 3.2 Calibration factor α

The calibration factor α was derived to scale $g_{cos}$ to match SiB4 COS plant assimilation with COS flux observations of

laboratory leaf gas exchange measurements (Berry et al. 2013). The $\alpha_{obs}$ values that we derived based on field measurements of COS ecosystem and soil fluxes, together with simulated $g_{cos}$, $g_s$ and $g_b$, are close to the value 1400 (Fig. 5), which support the initial calibration by Berry et al. (2013) using laboratory leaf gas exchange measurements. At the same time, however, we found $\alpha_{obs}$ to vary in time and between sites (Fig. 5), indicating that a single α value was not able to capture the variation of measured COS vegetation fluxes across sites and seasons. The average summer-time $\alpha_{obs}$ (June-August) of $1616 \pm 562$ was 15

% higher than the current value of 1400. This was consistent with our findings that, on average, SiB4 underestimates COS biosphere fluxes (Sect. 3.1.3). We did not find patterns in $\alpha_{obs}$ that apply to all PFTs in the same way that would have helped to update α in SiB4. However, for DBF and C3-GRA sites we observed that the 2-weekly average $\alpha_{obs}$ typically goes down with increasing air temperature for temperatures above ∼ 16 °C (Fig. S11). This observation requires further investigation from hourly data points and will be further discussed in our recommendations (Sect. 4.3).





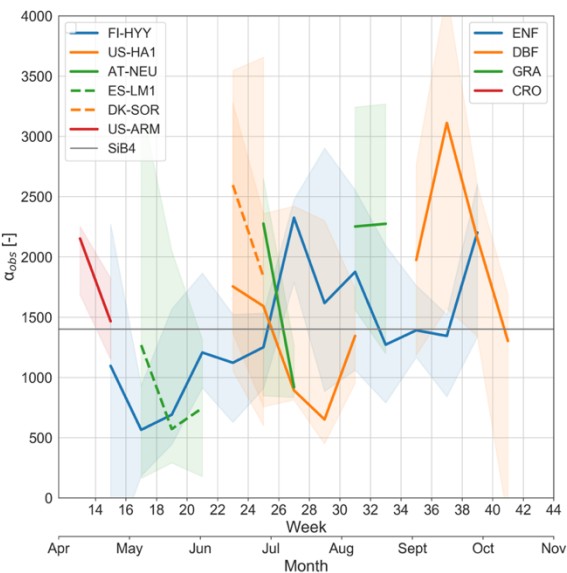

**Figure 5. Seasonal change of (2-weekly) median observation-based calibration factor α ($\alpha_{obs}$; see Eq. (10)) per site in which colors are separated by PFT. The shaded areas represent the 25th-75th percentiles.**

### 3.3 Global biospheric COS sink

The simulated global patterns in COS uptake were similar to that of GPP (not shown), due to the modeled vegetation COS uptake being coupled to GPP through the RuBisCO enzyme activity and stomatal conductance. Globally, the largest portion of COS uptake took place in tropical regions of South America, Africa, and Asia (Fig 6). In the Northern Hemisphere (NH), COS was mainly taken up during the summer months (Fig. 6b). The spatial distribution of COS uptake was also similar to that presented by Maignan et al. (2021) based on ORCHIDEE simulations. Using the original SiB4 model, i.e., the original Berry soil model and fixed 500 pmol mol$^{-1}$ COS mole fractions, the global COS biosphere sink amounts to $922 \pm 11$ Gg S yr$^{-1}$ over the years 2000-2020 with no substantial trend. 146 Gg S yr$^{-1}$ out of the total COS biosphere sink was taken up by the soil (Table 1). The change from the original Berry soil model to the Ogée soil model lowered the soil uptake in most regions globally (Fig. 7a, S12). The tropical soil COS uptake reduced from ~ 4-5 pmol m$^{-2}$ s$^{-1}$ to ~ 2-3 pmol m$^{-2}$ s$^{-1}$. In the NH, the soil uptake is also reduced due to the contributions of COS production in agricultural soils. The global COS soil sink thereby reduced from 146 to 104 Gg S yr$^{-1}$ when we changed from the original Berry soil model to the Ogée soil model, a 29 % reduction of soil uptake, but only a 5 % reduction of the total COS biosphere sink. The modification from a fixed COS mole fraction to spatially and temporally varying COS mole fractions caused an additional reduction of the global COS biosphere sink to 753 Gg S yr$^{-1}$ (Fig. 7b, S13). This 15 % reduction relative to a simulation with a constant and spatially uniform 500 pmol mol$^{-1}$ COS mole fraction illustrates the importance of accounting for varying COS mole fractions.



The largest drop in the global COS biosphere sink (169 Gg S yr⁻¹, i.e. from 922 to 753 Gg S yr⁻¹) occurs in the tropical regions (113 Gg S yr⁻¹ for latitudes between -23.5 and +23.5 °N) as the large biomass in that region leads to the largest COS uptake and the largest drop in COS mole fractions. This update is a significant contribution to solving the gap in the COS budget of ~ 432 Gg S yr⁻¹ (Ma et al., 2021); however, it does not fully eliminate the missing source in the COS budget. Ma et al. (2021) showed that assigning the missing COS source to the ocean yields more realistic distributions of COS mole fractions over the

tropical regions compared to assuming an overestimated biosphere sink. For these reasons, it is unlikely that the gap in the COS budget is solely caused by an overestimated tropical biosphere sink. Still, flux observations in the tropics would have to confirm this.

In Sect. 3.1.3 we found on average an 8 ± 27 % underestimation of the daytime COS vegetation flux as simulated by SiB4. If

we assume that the daytime uptake dominates the total COS uptake, and we correct the COS vegetation sink for the underestimation globally, then we find a vegetation sink of 717 ± 179 Gg S yr⁻¹ instead of 664 Gg S yr⁻¹, and a total biosphere uptake of 806 ± 179 Gg S yr⁻¹ instead of 753 Gg S yr⁻¹. Note, however, that this scaling is highly uncertain, because we found substantial variability between sites, and a large fraction of the uptake occurs in the tropics, for which we cannot validate the SiB4 model due to a lack of observations.

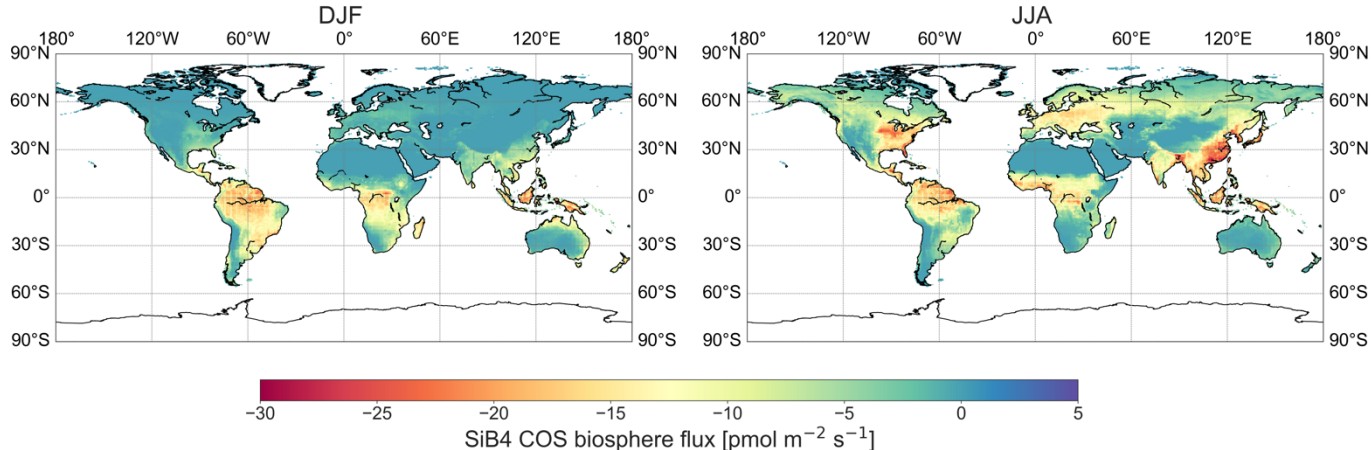


**Figure 6. Global distribution of the COS biosphere flux in winter (DJF, left) and summer months (JJA, right) as simulated by SiB4_var_Ogee over the years 2000-2020. Negative values indicate uptake of COS by the biosphere while positive values indicate COS emissions.**





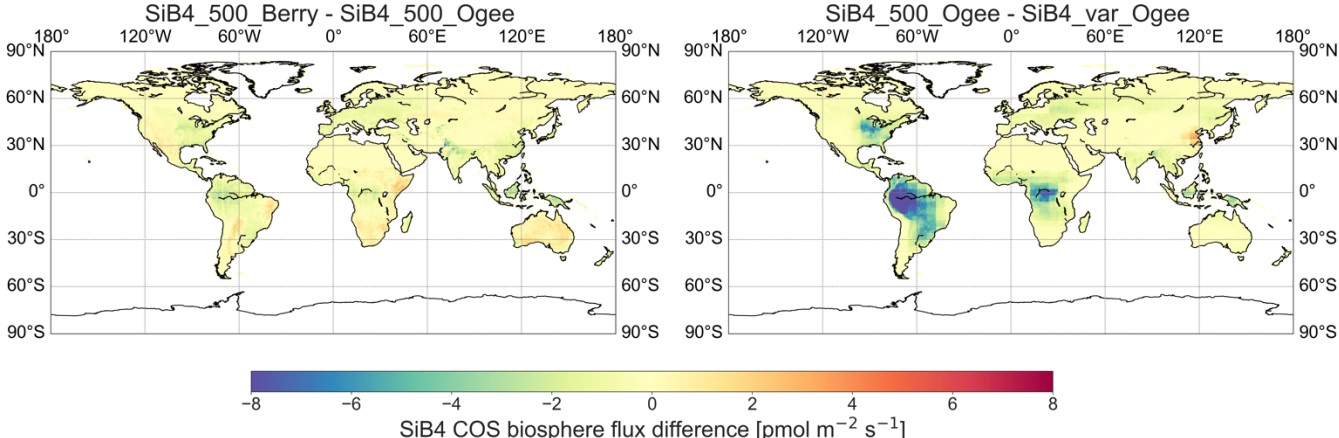

**Figure 7. COS biosphere flux difference between two SiB4 model runs. Left: difference between SiB4_500_Berry and SiB4_500_Ogee to show the flux difference between the soil models. Right: difference between the SiB4_500_Ogee and SiB4_var_Ogee to show the effect of changing to variable mole fractions. Negative values indicate a drop in the biosphere COS uptake.**

## 4. Recommendations for COS-specific future model development

We found model-observation biases that could be ascribed to different components of the model (depending on the site), such as the soil COS flux or vegetation COS uptake, where the latter was caused by underestimated enzyme activity that also links to GPP. If sufficient COS flux observations were available, these could help as an extra constraint to improve the model enzyme activity and thereby GPP. Such an approach would require a number of advancements in the understanding and implementation of COS biosphere exchange in SiB4. We have identified a number of ways to improve the COS flux

simulations in SiB4, which might also apply to mechanistic COS implementations in other biosphere models:

1.  Based on the analysis of the calibration factor $\alpha_{obs}$ (Sect. 3.2) **we suggest a refined calibration of the internal conductance $g_{cos}$ such that it captures the true temperature variation of COS vegetation fluxes**. The current calibration factor of 1400 is based on laboratory leaf gas exchange measurements of different plant species. However, our analysis based on field observations of COS vegetation fluxes shows that an alpha of 1400 mostly underestimates

the COS uptake, and suggest that the laboratory-based calibration did not fully resemble the COS vegetation uptake in the field. We suggest a re-calibration of $g_{cos}$ based on both laboratory and field observations of the COS vegetation uptake. Extra attention should be given to the temperature response of COS uptake. The modeled COS uptake is strongly coupled to GPP. However, several studies have shown that the ratio of COS to $CO_2$ deposition velocities (in literature also called the leaf relative uptake ratio) varies with temperature (Cochavi et al., 2021; Stimler et al., 2010)

and humidity (Sun et al. 2018; Kooijmans et al. 2019), in addition to the better known variability with light. The temperature response of the COS uptake is currently taken from $V_{max}$ and is scaled with an empirical temperature function (Eq. (2)) and an additional factor $T_{can}/T_0$ (Eq. (3)), where the latter increases the COS uptake at higher temperatures. However, the $T_{can}/T_0$ term has been added as a simple correction, but has not been empirically derived.




The temperature dependence of the CA enzyme activity could be determined from laboratory experiments, to be able to keep other effects (e.g. on mesophyll conductance) than temperature constant. Field observations could then be used to scale the laboratory-based calibration to ecosystem level and to different PFTs.

2.  The SiB4 model is capable of simulating nighttime COS vegetation uptake through stomatal opening, although the nighttime uptake was often underestimated (Fig. 3). **As nighttime COS vegetation uptake is driven by stomatal opening, the COS flux observations can be used to estimate nighttime (minimum) stomatal conductance values** (Berkelhammer et al., 2020; Maignan et al., 2021; Wehr et al., 2017). These values can be compared against those summarized by Lombardozzi et al. (2017) and tested in SiB4. However, **similar approaches and processing techniques are required to be able to evaluate the accuracy of the nighttime COS uptake and determine the nighttime stomatal conductance.** Changing the minimum stomatal conductance values would also have consequences for simulations of daytime carbon, water, and energy, which should also be (re-)evaluated.

3.  We have seen that the simulated COS soil flux can be very different depending on biome (in SiB4 selected as the PFT). This is especially true for fertilized soils that are typically found in agricultural sites, where large emissions of COS are observed. However, soils can contain high nitrogen contents regardless of whether or not it is an agricultural soil. Therefore, it is important to know the nitrogen content for setting the soil COS uptake and production parameter values for the COS soil flux calculation (Table 2). **We suggest the use of global maps of the soil nitrogen content and to use the relation between COS soil production and soil nitrogen content (Kaisermann et al. 2018b) for more accurate COS soil production simulations.**

4.  This study relied on the availability of field observations. We were able to evaluate SiB4 with the COS field observations available from a number of PFTs. **However, we lacked observations on evergreen broadleaf forests that are largely represented in tropical regions. Such observations could give further insights into the COS budget in tropical regions, where currently the largest uncertainties exist**. Moreover, controlled laboratory measurements of soil COS exchange have been shown to be very powerful to understand the soil COS exchange and to parameterize COS soil models (Meredith et al., 2018, 2019). However, field observations of COS soil exchange along with ecosystem COS fluxes are needed to evaluate COS soil models under field conditions (Ogée et al., 2016; Sun et al., 2015), which would also require standardization of measurement and processing techniques (Kohonen et al., 2020). Finally, the NOAA measurement network of atmospheric COS mole fractions has good coverage over North America and the Pacific Ocean, but other regions are less well represented. The COS mole fraction fields that we prescribed to the SiB4 model rely on the availability of COS observations. A better global coverage of COS mole fraction observations would therefore be beneficial, e.g. through the use of satellite data, where sensitivity to the middle and upper troposphere can currently be achieved (Glatthor et al. 2015; Kuai et al. 2014). Moreover, SiB4 should ideally be directly coupled to an atmospheric transport model to account for the interconnection between COS uptake and COS mole fractions.

**Biogeosciences** Open Access
Discussions
EGU

**Conclusion**

The experimental efforts made in the last decade to obtain field observations of COS ecosystem fluxes, now offer the possibility
of a unique SiB4 model validation of COS biosphere exchange over different biomes. SiB4 was demonstrated to be capable
of simulating the diurnal and seasonal variation of COS fluxes in the boreal, temperate and Mediterranean region, however
with an average underestimation of $8 \pm 27$ % of the daytime vegetation flux. The magnitude of the biases differed per site, but
could not be ascribed to a single component of the model.

We find a lower global soil COS sink with the implementation of the Ogée et al. (2016) soil COS model. Still, the soil COS
flux remains a relatively small component in the total COS budget, which benefits the use of COS as a global photosynthesis
tracer. A larger effect on the global COS biosphere sink was found by changing the fixed COS mole fraction of 500 pmol mol$^{-1}$
to values that vary spatially and temporally. The reduction in the COS sink strength is most pronounced in regions with large
biomass such as the tropics. This analysis highlights the importance of accounting for variations in atmospheric COS mole
fractions, which was not yet adopted as a standard practice.

We make a number of recommendations for future improvements of the model, including re-calibration of the COS model
parameters. However, we are limited by data coverage to be able to accurately constrain the model over different PFTs and
seasons. More campaigns and long-term observations in underrepresented PFTs, biomes and soil types would be key to
continued improvement of the model.

**Code and data availability**

The SiB4 code is available online at https://gitlab.com/kdhaynes/sib4_corral. SiB4 simulation output used in this study is
available at https://doi.org/10.5281/zenodo.5084644. COS campaign data are downloaded from the original data publications
as reported in Table 3. Long-term $CO_2$ flux timeseries from FLUXNET, AmeriFlux or ICOS are downloaded from the
references listed in Table S1.

**Author contributions**

LMJK and MK devised the study. LMJK and AC implemented the COS model developments with help from AK, IB, KH, JO,
LM, and WS. LMJK, AK, KH, IB, ITL, MG, WP and JM developed the procedure for running SiB4 simulations. LMJK
analysed the results with consultation of AC and MK. JM performed TM5 model inversion runs. WS, KMK, TV, IM, HC, FS,
GW, MB, MW, KM, US, RC, and RW provided data and site-specific insights. LMJK wrote the manuscript and all authors
provided comments.



**Competing interests**

The authors declare that they have no conflict of interest

**Acknowledgement**

We thank everyone that contributed to the collection of data through campaigns as well as the FLUXNET, AmeriFlux and ICOS network. Specifically, we acknowledge the Alexander von Humboldt Foundation for supporting the MANIP project with the Max-Planck Prize to Markus Reichstein that was used for the collection of ICOS data at ES-LM1. Data collection at FI-HYY was supported by ICOS-Finland (319871) and The Atmosphere and Climate Competence Center (ACCC) Flagship, funded by the Academy of Finland (grant number 337549). Data from the Sorø beech forest site (DK-SOR) have been

measured, evaluated and provided by Kim Pilegaard and Andreas Ibrom and the station team. The work was funded by the Technical University of Denmark (DTU), the Danish Research Council (DFF – 1323-00182), the Danish Ministry of higher Education and Science (5072-00008B) and the EU research infrastructure projects RINGO and ICOS. Operation of the US-HA1 site is supported by the AmeriFlux Management Project with funding by the U.S. Department of Energy's Office of Science under Contract No. DE-AC02-05CH11231, and additionally is a part of the Harvard Forest LTER site supported by

the National Science Foundation (DEB-1832210). Data collection at US-ARM was supported by the Office of Biological and Environmental Research of the US Department of Energy under contract No. DE-AC02-05CH11231 as part of the Atmospheric Radiation Measurement Program (ARM). We are very grateful to the principal investigators J. William Munger (US-HA1), Sebastien Biraud (US-ARM), Roser Matamala, David Cook (US-IB2), Tilden Meyers (US-BO1), Andreas Ibrom (DK-SOR) and Mirco Migliavacca (ES-LM1)

**Financial support**

This work was funded through the ERC-advanced funding scheme (AdG 2016 Project Number: 742798, Project Acronym: COS-OCS). SiB4 simulations were performed using a grant for computing time (17616) from NWO. TV was supported by the grant of the Tyumen region, Russia, Government in accordance with the Program of the World-Class West Siberian Interregional Scientific and Educational Center (National Project "Nauka"). IM was supported by ICOS Finland and ACCC

Flagship funded by the Academy of Finland grant number 337549. GW and FS received funding from the Austrian National Science Fund (FWF) through grants P27176, P31669 and I3859.

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
