# Peer review of "Evaluation of carbonyl sulfide biosphere exchange in the Simple Biosphere Model (SiB4)"

_Biogeosciences, 2021_

## Author Comment (AC1)

**We thank the reviewer for the comments and helpful suggestions. In the following we reply to the individual comments and clarify which modifications were made.**

**The reply is formatted as:** reviewer comment, author response, *revised text.*

**Review 1**

Linda M.J. Kooijmans and coauthors present an evaluation of the implementation of carbonyl sulphide (COS) fluxes in the terrestrial biosphere model SiB4. One can say without exaggeration that the earlier SiB4 implementation of COS is the reference for all current biosphere models that include COS. Kooijmans et al. present a very thorough evaluation with excellent supplementary information that answered almost all questions that arose while reading the manuscript.

My comments are hence minor.

1. I would disagree with the recommendation 4.1. α is not the only reason that COS fluxes are underestimated at some sites. GPP is also underestimated at DK-Sor and AT-Neu. The seasonal shape of GPP is very different in the model at US-IB2 compared to the estimated GPP from observations. So fitting α seems like a fudge factor. I would not recommend this.

> The reviewer makes a valid point, COS flux simulations are currently still linked to GPP simulations through $g_s$ and $V_{max}$ and therefore the accuracy of COS flux simulations go hand in hand with the accuracy of GPP simulations. We have removed the recommendation to adjust alpha. However, we would still like to keep the part of recommendation 4.1 where we emphasize the different temperature response of COS and CO2 uptake.

2. I also regret the wording in recommendation 4.2. The minimum stomatal conductance is called $g_0$ in the manuscript. It is not explained how it is used in the model. Stomatal conductance most often depends on net assimilation in conductance formulations such as Ball-Berry and its variants. Net assimilation is negative during dawn and dusk. Is stomatal conductance then set to $g_0$? This is questionable. If I remember well, Ball et al. (1987) said that $g_0$ is simply the fitted intercept in the empirical formulation during daytime photosynthesis. It is not the nighttime value. If ever I would recommend to look into such formulations as in Barbour and Buckley (PCE 2007).

> We thank the reviewer for pointing out the differences between $g_0$ and the nighttime conductance (which we now call $g_{dark}$). We have indicated these differences now in section 3.1.3 and the recommendation in 4.2. Still, we assume that $g_0$ is representative of $g_{dark}$, which is a common assumption in land surface models (e.g. Lombardozzi et al., 2017). This is now also explicitly mentioned in the text.

3. I was missing the explanation/discussion that the authors used reanalysis data to drive the model and not local observations. But especially the discussion of the underestimation of GPP and COS fluxes at DK-Sor literally screamed for it. Would it be possible to redo say SiB4_var_Ogee of Figure 2 using local meteo?

The measurement periods of the COS campaigns are not always covered by the FLUXNET, ICOS or AmeriFlux datasets to be used as meteorological driver input. Therefore, we run the SiB4 simulations with MERRA driver data as it provides consistency in data collection, availability and application across sites. We make a note of this in section 2.1.4.

Still, based on the reviewers' comments we also added a comparison of SiB4 runs with MERRA meteorology and observed site meteorology as driver input for two sites that have more than 10 years of observations and where the site meteorology covers the COS measurement period (FI-HYY and DK-SOR) in the supplement (Fig. S7, and shown below). The comparison for DK-SOR shows that the SiB4 run with site meteorology provides a similar drought anomaly in 2016 as the run with MERRA meteorology. The fact that SiB4 is not able to capture the GPP anomaly is thus not due to the driver data used. This is now mentioned in section 3.1.3.

We do also find that GPP is consistently lower with observed site meteorology due to lower PAR in the observations than in MERRA, leading to a larger model-observation bias with the site meteorology. A reason for the larger bias with site meteorology is that all of the tests, development and tuning for SiB4 are done with MERRA driver files, which might be different for site-level input. We made a note of this in the supplement:

*Simulated GPP is consistently lower in SiB4_Obs due to lower radiation in the observations than in MERRA2 (Fig. S7c,g), leading to a larger GPP model-observation bias with SiB4_Obs (Fig. S7a,e). A reason for the larger bias when site meteorology is used is that all of the tests, development and tuning for SiB4 are done with MERRA2 driver files, which might be different for site-level meteorological input. The results in the main text are consistently based on SiB4 runs with MERRA2 driver data, and are not biased by using different meteorology than in the SiB4 development.*

We also must note that we found that for a few sites we accidentally selected the wrong year Figs. S10 (Tsoil) and S11 (SWC) (i.e., 2015 instead of 2016 for FI-HYY, DK-SOR, ES-LM1 and US-IB2. This made it seem like the MERRA data did not resemble the observed conditions in Figs. S9 and S10 (SWC). This mistake is now corrected.

[Figure]

**Figure S7: Seasonal cycles of GPP, canopy temperature (Tcanopy), photosynthetically active radiation (PAR) and leaf area index (LAI) at DK-SOR (top) and FI-HYY (bottom) as simulated by SiB4 with MERRA2 driver data (blue) and as simulated by SiB4 with observed site meteorology (orange) and compared with GPP observations from ICOS.**

4. I would have loved to see the comparison of SiB4 output with the inverted fields of Ma et al. (2021).

> We have added the figure below to the supplementary material and discuss it briefly in section 3.3:
>
> *The biosphere flux resulting from inverse modelling by Ma et al. (2021) indicates COS emissions in the Amazon (Fig. S17). While biosphere emissions over the Amazon are unrealistic (Glatthor et al., 2015), it reflects the large missing source in the Tropics (land and ocean) that we are not able to attribute to the biosphere; see a comparison of our SiB4 biosphere flux with the inverted biosphere flux by Ma et al. (2021) in Fig. S17. A potential reason for unrealistic attribution of missing sources of COS is that there are no NOAA observations in the Tropics and its upwind regions to constrain the TM5 inversions.*

[Figure]

**Figure S17. Global distribution of the COS vegetation flux as simulated by SiB4_var_Ogee (left) and the posterior biosphere flux as presented by Ma et al. (2021).**

5. I think that Figure 1 is redundant given Table 3 and I would remove it.

We have moved Figure 1 to supplementary information.

6. I found the notation $V_{max}$ pretty unusual. I had to go back several times to Equations 2 and 3 to check the definition. I would recommend to use something like $V_{c,max25}$, which is pretty standard and tells the important information, i.e. it is for carboxylation and at 25 °C.

Corrected as suggested

**References**

Glatthor, N., Höpfner, M., Baker, I. T., Berry, J., Campbell, J. E., Kawa, S. R., Krysztofiak, G., Leyser, A., Sinnhuber, B. M., Stiller, G. P., Stinecipher, J. and Von Clarmann, T.: Tropical sources and sinks of carbonyl sulfide observed from space, Geophys. Res. Lett., 42(22), 10082–10090, doi:10.1002/2015GL066293, 2015.

---

## Author Comment (AC2)

**We thank the reviewer for the comments and helpful suggestions. In the following we reply to the individual comments and clarify which modifications were made.**

**The reply is formatted as:** reviewer comment, author response, *revised text.*

**Review 2**

The paper presents an update of the modelling of the carbonyl sulfide biosphere fluxes within the SiB4 model. The soil model of Ogée et al. (2016) is implemented and a spatially and temporally varying COS atmospheric concentration is considered. This latter modification is shown to have a large impact. The new models are evaluated against field observations and a revised budget for COS is given at global scale, reducing the missing source. The authors make valuable recommendations, with a fine study on the alpha parameter of the COS vegetation uptake model.

The paper is well built and well written, with clear figures, and represents an important new contribution to this research field linking COS uptake and GPP.

I have only minor comments and some requests for clarification.

Page 2, line 65: "the Lund-Potsdam-Jena model (LPJ) and the Community Land Model (CLM4) (Launois et al. 2015a)" -> Launois et al. cannot be a reference for a possible implementation of biosphere COS exchange for LPJ and CLM as, as you further state in Table 1, they only scaled the GPP using a leaf relative uptake approach to estimate vegetation COS fluxes.

> We thank the reviewer for pointing this out and have removed the references to LPJ and CLM4 from the text.

Page 3, line 95: Vesala et al., in prep -> Vesala et al., 2021. They mention that "e is the original e multiplied by 2.1, the average ratio of Hyytiälä and SiB4 LAI data". How do you reconcile having a factor 2 on LAI with however correct simulations of GPP (Figure S3a) and COS fluxes (Figure 2a)?

> The difference originates in a different LAI definition; that is, Vesala et al. (2021) report the all-sided leaf area, SiB4 on the other hand defines LAI as one-sided LAI, which explains the factor 2. This will be further clarified in Vesala et al. (2021).

Page 7, lines 177-178: "These numbers were later updated to alpha = 1400 and 8862 for C3 and C4 species, respectively, after updates were made to the SiB model." -> Could you explain a bit more what were these updates and whether these parameters were recalibrated against measurements?

> The alpha values were modified in SiB3 after reanalysis of the gas exchange data (Stimler et al. 2010, 2011). We have now specified this in the text:
> *These numbers were later updated to $\alpha$ = 1400 and 8862 for C3 and C4 species, respectively after reanalysis of the gas exchange data.*

Page 8, lines 198-199: "These effects of nutrient fertilization on soil COS exchange were initially not simulated in the SiB4 model." -> Do you mean they are simulated now?

The current implementation using the soil model by Ogee et al. (2016) indeed includes a higher COS soil production and lower uptake rates for agricultural soils by using biome-specific production- and uptake parameter values that are related to the plant functional types (Table 2). The implementation does not include simulations of nutrients (i.e. nitrogen, phosphorus) itself. We have clarified more explicitly at the end of section 2.1.2 how fertilization effects are now simulated in the current implementation:

*Here, the $f_{CA}$ for agricultural soils is substantially smaller than that of other vegetated biome types, thereby including the reduced COS uptake in fertilized (agricultural) soils (Kaisermann et al. 2018b).*

and

*The higher value of a for agricultural soils (Table 2) allows for higher COS soil production in this soil type.*

Page 9, lines 226-228: "We chose the tortuosity functions of Deepagoda et al. (2011) for air and Millington and Quirk (1961) for water, as these functions do not require a pore-size distribution parameter, which facilitates its implementation in SiB4." -> Do you mean you have a constant porosity per soil type?

The soil porosity used in SiB4 varies per grid cell, taken from a soil characteristics database (see section 2.1.4). What we mean here is that the pore-size distribution parameter is not part of the tortuosity functions of Deepagoda et al. (2011) and Millington and Quirk (1961). To clarify this we have changed the sentence to:

*We chose the tortuosity functions of Deepagoda et al. (2011) for air and Millington and Quirk (1961) for water, as these functions do not **depend on** a pore-size distribution parameter, which facilitates its implementation in SiB4.*

Page 9, line 233: "kuncat varies with soil pH" -> How is soil pH prescribed/computed in SiB4?

The soil pH is not specifically considered in SiB4 and we keep the pH constant at 4.5. We have added this information to section 2.1.2.

Page 9, equation (6): Is the production term valid for both oxic and anoxic soils?

Unfortunately SiB4 does not discriminate between oxic and anoxic (wetland) soils. Ideally we would include a "wetland" biome classification that takes into account the typically large production that has been observed in wetland soils (Whelan et al., 2013; Meredith et al., 2018). However, we were not able to implement this as wetlands are not considered in SiB4. We added this information in section 2.1.2:

*Ideally, we would include production parameter values for wetland soils so that we take into account the typically large production that has been observed in wetland soils (Whelan et al., 2013; Meredith et al., 2018). However, SiB4 does not discriminate between oxic and anoxic (wetland) soils, which precluded the implementation of wetland-specific COS soil production.*

Page 11, line 284: "CO2 mole fractions were held constant at 370 μmol mol-1 during spinup and simulations" - > Why don't you at least use varying annual means to get the CO2 fertilization effect in the simulations?

We have recently tested several CO2 mole fraction settings, including the option to vary CO2 over time. However, it is still quite inconclusive how assimilation would respond to CO2 fertilization in reality. For example, observations suggest that CO2 assimilation may not change, while stomatal conductance decreases, leading to enhanced water use efficiency (Brienen et al., 2011). Our first tests with increasing CO2 mole fraction settings in SiB4 showed large net carbon sinks in the tropics while observations suggest that the Amazon is not a large net carbon sink currently (Gatti et al., 2021). We have therefore decided to work with a constant CO2 mole fraction (biosphere in steady state) while we are still working on an improved implementation of the CO2 fertilization effect in SiB4. We added the following sentence to the section:

*Research is ongoing to implement an accurate representation of the effect of $CO_2$ fertilization in SiB4.*

Page 11, lines 289-290: "To compare SiB4 with site observations (listed in Table 3), we run the SiB4 model with 3-hourly output for only the grid cells (at 0.5° x 0.5° resolution) in which the sites are located" -> Why didn't you use the local meteorology available from the FLUXNET, ICOS or AmeriFlux sites? This comment is also valid for your remark Page 15, lines 384-388 that the model temperature (from MERRA?) in 2012 was higher than the observed one at the US-ARM site.

A similar comment was made by reviewer 1. We agree that using local meteorology in SiB4 would be ideal for comparison with observations. However, the measurement periods of the COS campaigns are not always covered by the FLUXNET, ICOS or AmeriFlux datasets to be used as meteorological driver input. Therefore, we run the SiB4 simulations with MERRA driver data as it provides consistency in data collection, availability and application across sites. We make a note of this in section 2.1.4.

Still, based on the reviewers' comments we also added a comparison of SiB4 runs with MERRA meteorology and observed site meteorology as driver input for two sites that have more than 10 years of observations and where the site meteorology covers the COS measurement period (FI-HYY and DK-SOR). These figures are now shown in the supplement (Fig. S7) and below. The comparison for DK-SOR shows that the SiB4 run with site meteorology provides a similar drought anomaly in 2016 as the run with MERRA meteorology. The fact that SiB4 is not able to capture the GPP anomaly at DK-SOR is thus not due to the driver data used. This is now mentioned in section 3.1.3.

We also find that GPP is consistently lower with observed site meteorology (Fig. S7a, e) due to lower PAR in the observations than in MERRA (Fig. S7c,g), leading to a larger model-observation bias with the site meteorology. A reason for the larger bias with site meteorology is that all of the tests, development and tuning for SiB4 are

done with MERRA driver files, which might be different for site-level input. We made a note of this in the supplement:

*Simulated GPP is consistently lower in SiB4_Obs due to lower radiation in the observations than in MERRA2 (Fig. S7c,g), leading to a larger GPP model-observation bias with SiB4_Obs (Fig. S7a,e). A reason for the larger bias when site meteorology is used is that all of the tests, development and tuning for SiB4 are done with MERRA2 driver files, which might be different for site-level meteorological input. The results in the main text are consistently based on SiB4 runs with MERRA2 driver data, and are not biased by using different meteorology than in the SiB4 development.*

[Figure]

**Figure S7: Seasonal cycles of GPP, canopy temperature (Tcanopy), photosynthetically active radiation (PAR) and leaf area index (LAI) at DK-SOR (top) and FI-HYY (bottom) as simulated by SiB4 with MERRA2 driver data (blue) and as simulated by SiB4 with observed site meteorology (orange) and compared with GPP observations from ICOS.**

On the same order, this means you are using a soil type defined at a 0.5° spatial resolution, what if the local soil type is different?

We do discriminate between different soil/biome types for uptake and production parameter values (Table 2), such as agricultural and non-agricultural soils. Further optimizing soil conditions such as soil porosity and pH would be possible, but we chose to test the quality of SiB4 simulations with the default SiB4 input, which helps explain biases from inversions on the global scale. A more site-specific analysis such as is done by Sun et al. (2015) would be interesting, but would require sufficient information on local soil conditions, which are often not documented (e.g. in the case of porosity), or vary even in the vicinity of the observation site.

Page 12, lines 316-317: "Ecosystem fluxes are corrected for storage of COS in the canopy airspace using collocated canopy COS profile measurements when available (FI-HYY and US-HA1)." -> Can you explain this a bit more?

We now explain the storage flux in more detail including a reference for more methodological details:

*The ecosystem fluxes determined by the EC technique can be biased due to storage (typically depletion) of COS in the canopy airspace under limited turbulent mixing. The air depleted in COS can then suddenly be captured by the EC system when turbulence is enhanced in the morning. Ecosystem fluxes therefore ideally need to be corrected for such storage change. We corrected the ecosystem fluxes for storage of COS in the canopy airspace using collocated canopy COS profile measurements when available (FI-HYY and US-HA1). More details on the storage flux calculation can be found in Kooijmans et al. (2017).*

Page 21, lines 544-545: "922 ± 11 Gg S yr-1 over the years 2000-2020" -> How do you compute this uncertainty? Is that the interannual variability?

The uncertainty here is calculated as the standard deviation over the years 2000-2020. We added this for clarification.

Figure S7: The diurnal cycles of COS soil fluxes seem to be often in opposite phase between observed and simulated (for ES-LM1, AT-NEU notably). Do you have an idea why?

The diurnal cycles of ES-LM1 and AT-NEU in Fig. S7 were representing the C3-GRA plant functional type and so did not show large COS soil production. However, when these two sites are simulated as agricultural soil (with different production parameter values for this soil type, see Table 2) we do find larger COS production. This larger COS production is more consistent with observations as was also explained in the main text. We have added the runs representing agricultural soils for ES-LM1 and AT-NEU to Fig. S7 (now Fig. S9). The updated results show smaller uptake and often net emissions due to COS soil production. The clear opposite phase that was visible when the sites were simulated as C3-GRA is now hardly present with the simulation representing agricultural soil.

**References**

Brienen, R.J.W., Wanek, W. & Hietz, P. Stable carbon isotopes in tree rings indicate improved water use efficiency and drought responses of a tropical dry forest tree species. *Trees* **25,** 103–113 (2011). https://doi.org/10.1007/s00468-010-0474-1

Gatti, L.V., Basso, L.S., Miller, J.B. *et al.* Amazonia as a carbon source linked to deforestation and climate change. *Nature* **595,** 388–393 (2021). https://doi.org/10.1038/s41586-021-03629-6

Whelan, M. E., Min, D.-H., and Rhew, R. C.: Salt marshes as a source of atmospheric carbonyl sulfide, Atmos. Environ., 73, 131–137, 2013.